# Comparison of Recently Proposed Causes of Climate Change

Stuart A. Harris 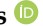

Department of Geography, University of Calgary, Calgary, AB T2N 1N4, Canada; harriss@ucalgary.ca

**Abstract:** This paper compares the ideas contained in the main papers published on climate change since World War II to arrive at a suggested consensus of our present knowledge regarding climatic changes and their causes. Atmospheric carbon dioxide is only suggested as a cause in one theory, which, despite its wide acceptance by Politicians, the media, and the Public, ignores the findings in other studies, including the ideas found in the Milankovitch Cycles. It also does not explain the well-known NASA map of the changes between the global 1951–1978 and the 2010–2019 mean annual temperatures. The other theories by Oceanographers, Earth scientists, and Geographers fit together to indicate that the variations in climate are the result of differential solar heating of the Earth, resulting in a series of processes redistributing the heat to produce a more uniform range of climates around the surface of the Earth. Key factors are the shape of the Earth and the Milankovitch Cycles, the distribution of land and water bodies, the differences between heating land and water, ocean currents and gateways, air masses, and hurricanes. Low atmospheric carbon dioxide levels during cold events could result in too little of this gas to support photosynthesis in plants, resulting in the extermination of most life on Earth as we know it. The 23 ka Milankovitch cycle has begun to reduce the winter insolation received at the surface of the atmosphere in the mid-latitudes of the Northern Hemisphere starting in 2020. This results in extreme weather as the winter insolation reaching the surface of the atmosphere in the higher latitudes of the Northern Hemisphere decreases while the summer air temperatures increase. It heralds the start of the next glaciation. A brief outline is given of some of the climatic changes and consequences that may be expected in western Canada during the next 11.5 ka.

**Keywords:** climate change; expected future weather in Western North America; glacial advances; Milankovitch Cycles; implications for future energy sources

## 1. Introduction

Since the biota first developed on the surface of the Earth, climate has always been critical to their survival. This has resulted in numerous examples of stories around catastrophic events being passed down by humans from generation to generation dealing with extreme weather, e.g., the Sagas of the Icelandic peoples, Noah's Ark, and in the Holy Bible. Major advances were made during the 19th century, including the development of geological mapping, which involved the interpretation of past environments under which the rocks were deposited in southern England, together with the first estimates of their age. Exploration of the world had reached a stage where French and German climatologists were able to develop good maps of the climate of the world together with an acceptable classification that is still widely used [1]. Gradually, the processes involved in erosion were sorted out; e.g., the glacial theory of Karl Schrimper and Louis Agassiz replaced part of the diluvial theory that had held sway for decades [2].

One of the key developments was the realization of the role of Earth-Sun relationships. There is one special group of cycles that are critical in determining the amount of solar radiation arriving at a given location on the surface of the Earth over long periods of time, viz., the Milankovitch cycles, first thought of by Adhemar (Croll [3,4]). Milankovitch [5,6] refined the calculations of the effect of the three kinds of Earth orbital movements that can alter the Sun's incoming radiation by up to 25% in the subtropical zones (30–60° north and

south of the equator). They are the shape of the Earth's orbit (eccentricity, a 100,000-year cycle), the angle of tilt of the Earth's axis relative to the Earth's orbital plane (obliquity, from 21.1–24.5° in a 41,000-year cycle), and the direction in which the Earth's axis of rotation is pointing (precession, a 23,000-year cycle). Milankovitch calculated that cold events might occur approximately every 41,000 years. Subsequent work shows that these are key controls affecting the climate of the Earth.

To make progress in Climatology, it has been shown that a good, reliable database of the constituent properties of the climate is necessary to obtain a reasonable average for that time slice. The difficulty is coping with the tremendous variation in conditions over the entire Earth at all time scales. The data should also be complete and obtained by a consistent method of measurement for all stations. Ideally, there should be no breaks in the data, although this is not usually achieved. Changes in methodology, equipment, and storage of large volumes of data have been major problems. As a result, most scientists studying the subject work with models of meteorology and weather forecasting.

This paper will examine the new theories concerning the causes and mechanisms of climate change that have been suggested since World War II. They are quite revolutionary and make use of the more recent developments in instrumentation, new discoveries by oceanographers, continuous monitoring of environmental factors, and the fast-growing data bank using up-to-date methods. However, first it is necessary to summarize the recent discoveries regarding the mechanism of global climatic change.

## 2. New Data on the Mechanism of Global Warming

It is now known that solar radiation supplies more than 99.95% of the total energy driving the world's climate [7]. The fact that the bulk of the solar radiation arrives on the surface of the Earth along the zone between the Tropics of Capricorn and Cancer, decreasing towards the Poles, results in a tremendous imbalance of initial heat distribution around the globe. The amount of solar heating at the polar latitudes throughout the year varies greatly, with the polar latitudes receiving considerably more solar energy in the summer than in the winter, when they receive no solar heat at all. As a result, in the winter hemisphere, the difference in solar heating between the equator and that pole is very large. This causes the large-scale circulation patterns observed in the atmosphere in the northern hemisphere. The difference in solar heating between day and night also drives the strong diurnal cycle of surface temperature over land.

### 2.1. Thermal Properties of the Earth's Surface

A total of 70% of the Earth's surface consists of water, with the remainder being land (rock, soil, or ice). The albedo of ice ranges from 0.5 to 0.7, so ice- and snow-covered surfaces reflect much of the incoming solar radiation back into space. Water has a very high heat capacity (4.187 mJ/m$^3$ K), so it can store or transport large quantities of heat in a given volume of water [8]. In addition, it absorbs over five times as much heat as soil or rock since it is translucent [9,10]. Currents, convection, and wave action mix the water, whereas transmission into a rock or sediment must be by conduction. Reradiation only occurs in the surface layer (water or land).

### 2.2. Transport of Heat towards the Poles

Dry air has a low heat capacity, but air can carry moisture in the form of water vapor, water droplets, or snow. Where water droplets are involved, the quantity of water carried can be enormous in Monsoons and Hurricanes. Accordingly, warm ocean currents and Hurricanes are the main carriers of heat from the Tropics towards the polar regions [8]. There can also be "rivers of water" carried to land areas by Monsoons in subtropical areas.

The warm ocean currents carry large quantities of heat towards the Poles but are constrained by the distribution of land and water (Figure 1).

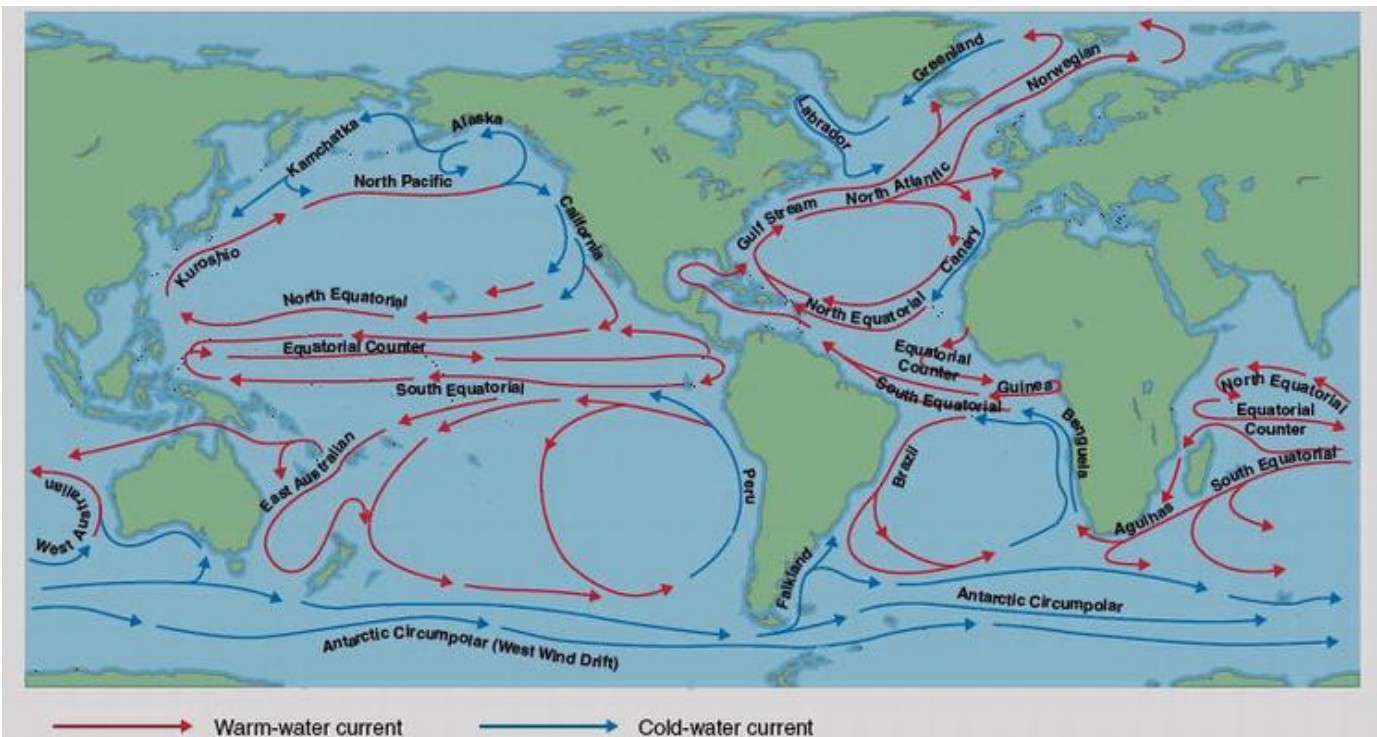

**Figure 1.** Distribution of cold and warm currents around the world [11]. Note that the warm currents are prevented from warming Antarctica by the cold Antarctic Circumpolar current, whereas the warm Gulf Stream penetrates northward into the North Atlantic Subarctic waters.

Gaps between continents form gateways, which aid in the movement of heat [12]. In the southern hemisphere, the ice cap is circular apart from the Falkland Islands archipelago, so the heat cannot be transported there, whereas the Atlantic Ocean provides a pathway for the Gulf Stream to warm the surrounding lands. Since the Earth rotates from east to west, the air masses move at varying speeds eastward in the form of waves (Rossby Waves) with fronts where warm and cold air interact, producing precipitation [13]. The Kuroshio current in the North Pacific Ocean is similar except that it is deflected westward when colliding with the cold glacial meltwater coming from the Wrangell-St. Elias Range of South Alaska. The shallow water (<50 m) in the Bering Straight also limits the water's movement. Hurricanes also transport large amounts of heat northward but are largely absent in the southern hemisphere.

### 2.3. Sources of Cold Air Masses

The primary source of old, dense Arctic Air is in the interior valleys of the mountains in Northern Siberia, where the coldest air temperatures commonly exceed −65 °C in winter [12]. They are partly fed by cold air drainage from Tibet flowing down its northern slope to the Hexi Corridor, and similar cold temperatures have been recorded from Fort Nelson, British Columbia [14], and from the high mountains in Utah. The cold Siberian air moves eastward along three main paths (Figure 2) and results in several different patterns of ice caps during the Wisconsin glaciation in North America. Path I is mainly used during the initial growth of the ice sheets and for changing Arctic air to Subtropical air, bringing about deglaciation of the western ice sheets during the retreat of the ice from its maximum glacial extent and the beginning of the subsequent Interglacial event.

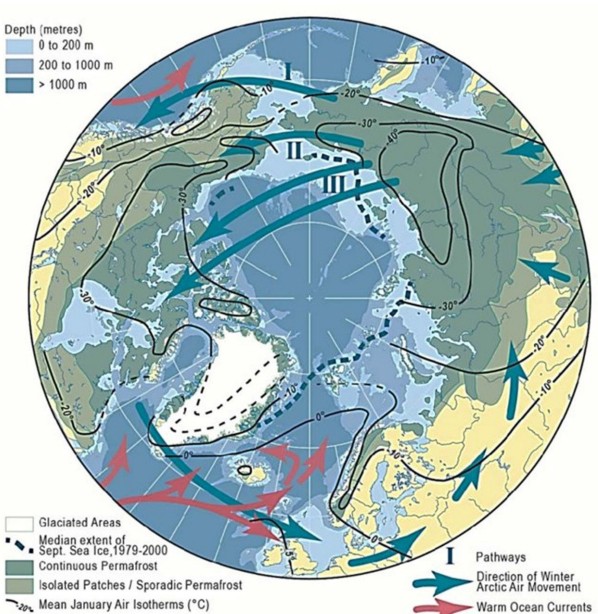

**Figure 2.** Map showing the distribution of permafrost in the Arctic together with the mean surface air January isotherms (°C) and the adjacent warm and cold currents [13]. Note the three main paths (I to III) taken by the Arctic air as it moves from Siberia to northern Canada and the positions of the main warm ocean currents bringing heat from the Tropics.

The coldest temperatures reported to date come from Antarctica, where there was a sudden decrease in winter temperatures in 2018 to set a new cold daily record of −98 °C [15]. Due to its being roughly circular and located over the South Pole, the main ice cap remains cold throughout the year and is surrounded by an ice-cold sea. The circum-Antarctic current acts as a buffer between it and the other southern continents.

*2.4. Sources of Warm Air Masses*

Over oceans, this is primarily centered in the zone of Intertropical Convergence between the Tropics of Cancer and Capricorn, where the sun is overhead for part of the year [10] (Figure 1). These are the main sources of the warm ocean currents that transfer heat northward in the northern hemisphere from the main hot centers of the oceans. Overheating of the tropical waters also results in onshore Monsoon events and "rivers of rain" coming northeastward from the Indian Ocean and the Central Pacific Ocean that bring large quantities of moisture to Subtropical areas such as India and southwestern North America.

For large areas of land, warm air masses originate where large dry deserts exist, such as the Sahara Desert in Africa, the Mohave Desert in Arizona, North America, or the Interior of Australia. The effects of these expand and contract as the sun changes position during the year and can bring drought conditions to southeast Africa and southern Europe.

*2.5. Effects of Humans on Climate Change*

The IPCC argues that carbon dioxide coming from industrial plants controls the air temperature [16,17] (see Section 3.5). Certainly, deforestation, logging, agriculture, and urbanization have altered the albedo on land, but these changes do not produce sufficiently large temperature changes to be significant when compared with the quantity of solar radiation reaching the surface of the Earth. They may, however, cause substantial changes in precipitation, as in the case of Costa Rica, where deforestation of 85% of the rain forest resulted in a reduction in precipitation of c.30%. There is a marked difference between the warming of cities by the heat island effect and the rural areas of the northern hemisphere, which have not shown marked warming during the last 10 years [18–21].

## 3. Post-World War II Theories of Causes of Climate Change

There were immense technological advances during World War II, and the countries of the world had to cooperate in solving problems, sharing technologies, and generally learning more about our natural environment, especially climate. Oceanographic surveys were conducted and produced very valuable information about the sea floor. This complemented an explosion in studies of the landscape and resulted in the development of new ideas regarding the causes and mechanisms of climate change.

### 3.1. Identification of Cold Events on Land

Despite numerous studies, no complete sequence of cold and warm events could be found in sections of glacial deposits that provided a complete record of climatic changes during the late Cenozoic and Pleistocene Periods. There has been too much interference and periodic erosion of glaciers and sediments to allow for a complete record to be preserved. The best records come from inland drainage basins in Columbia, but even then, the long time span representing the entire period of colder climates means that little detail can be obtained from even thin layers. The use of radiometric dating showed that there were big gaps in most sequences, and past correlations of glacial deposits across the Great Plains in North America were often not reliable [22–24].

The problem was partly solved by making a histogram of all the available dated glacial deposits and associated permafrost features in North America [25]. This provides a list of the 13 best-preserved major deposits ranging back to 3.5 Ma B.P., and these could be correlated with some marine transgressions with warmer faunas along coastal Alaska [26]. While the list was not complete, it provided a first look at the length of the past glaciations on that continent. A more complete list covering all the glaciers in the world has subsequently been published [27].

Magnetostratigraphic studies have established that the pattern of areas of ice sheets in North America changed each time there was a major change in the direction of the polarity of the Earth [24,25]. This seems to be correlated with changes in the topography at the time of the reversal of the Earth's magnetic field. Thus, the pattern of cold cycles during the last eight glaciations, each spanning 100 ka, was similar [26], although the extent of the ice sheets varied somewhat from one event to another [24,25].

### 3.2. Identification of Cold Events in the Oceans

Some of the most important evidence for climatic changes has been found by Oceanographers. These include fluctuations in sea temperatures in the deep-sea cores and evidence for the transport of solar heat from the equatorial areas by warm currents in the seas and by hurricanes, as well as by deep thermohaline currents.

### 3.3. Fluctuations in Sea Temperatures Measured by $\delta O^{18}$ in Foraminifera

Shackleton was the first to report numerous alternating warm and cool assemblages of layers from deep sea cores in the Atlantic Ocean. Subsequent work showed that there were over 100 such fluctuations in the last 3.3 Ma B.P., and these became more marked in the upper layers of the cores, while the amplitude of temperature fluctuations increased towards the sediment surface (Figure 3) [28–32]. They showed a progressive cooling of the North Atlantic Ocean beginning about 3.5 Ma B.P. [27]. However, the frequency of the cold peaks is much greater than the 41 ka calculated by Milankovitch and appears to be controlled by his 23 ka precession cycle. The 41 ka cycle must be part of the cause of the variation in degree of cold from one cold period to the next.

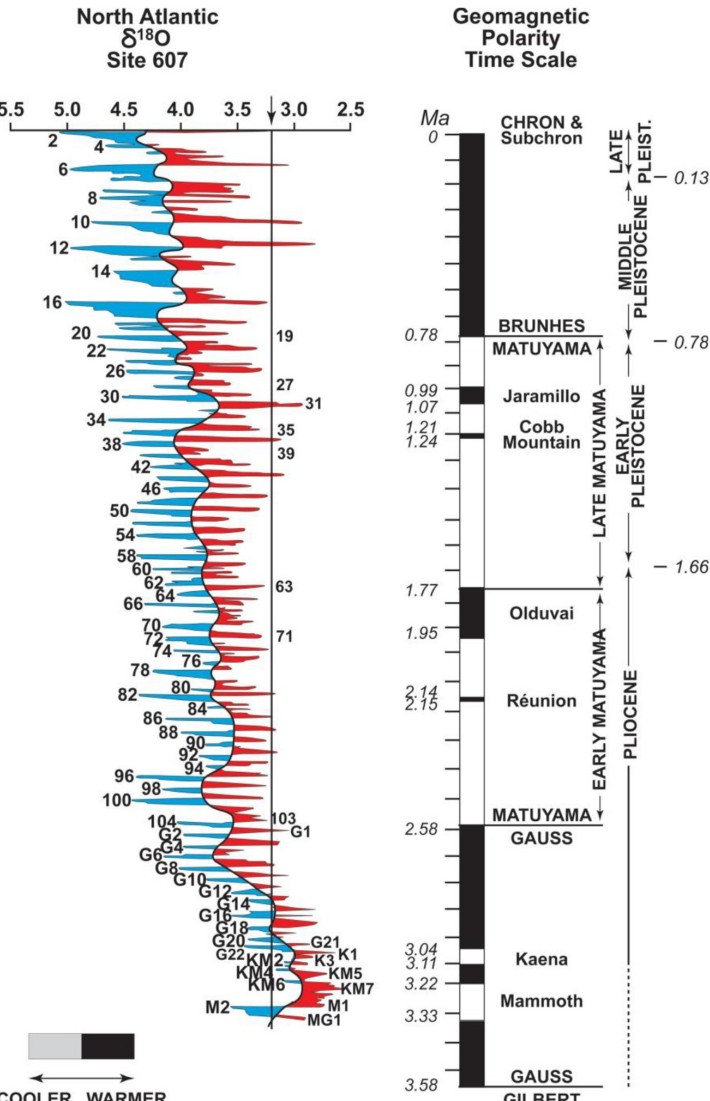

**Figure 3.** Oxygen isotope palaeotemperature record [27–29] and geomagnetic polarity timescale [24,25]. Black and white areas are normal and reversed polarity, respectively. The arrow at the top indicates the mean Holocene oxygen isotope value. Numbers on the peaks and troughs are the isotope stages (modified from [30]).

For convenience, the peaks and troughs were numbered starting with 1 for the recent Interglacial [33]. These peaks have been widely used to label climatic changes in the areas around Northwest Europe, and it is often assumed that these changes were world-wide in extent. Unfortunately, it has since been shown that different parts of Canada had quite different climatic histories during the Wisconsin Glaciation [34], and it has become obvious that there is considerable variation in climate depending on the geographical environment [35] and the timing relative to the 41 ka cycle.

The buildup of heat in the North Atlantic warms the cold Arctic air moving eastward towards western Europe and leads to the thawing of the ice cover in the Arctic Ocean. Cold, dry Siberian air moving eastward across this open water picks up both heat and water vapor, leaving behind relatively saline, warm water that has a higher density. When the density differential is large enough, the warm, saline water sinks to the bottom of the ocean, where it accumulates. It is estimated that at least 60% of the heat accumulated in the North Atlantic area since the end of the last Neoglacial event (c. 1915 A. D.) is held in the sea water. (Figure 4).

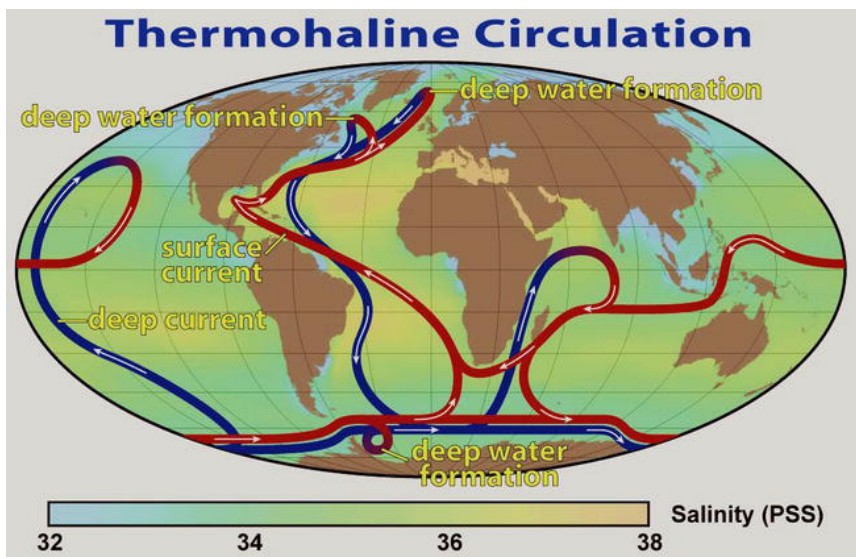

**Figure 4.** Paths of the thermohaline deep water circulation of warm North Atlantic bottom water (red) and the cold Subarctic return flow (blue) around the oceans [13].

*3.4. The Deep-Water Thermohaline Currents*

Inevitably, this buildup of heat in the North Atlantic creates a situation that results in periodic drastic events in the oceans. Oceanographers discovered a deep-water thermohaline circulation system over 50 years ago [30,36–42], although they did not speculate on the source of the heat. They have carried out enough research to demonstrate that there is a cycle of climate change that has been occurring every 100 ka during the last 800 ka B.P. It consists of fast-moving, deep thermohaline currents (THC) that move heat down to the southern hemisphere and form part of a global thermohaline system [40]. A return flow of cold Antarctic surface water moves north to the North Atlantic Ocean to replace it and restore the former sea level [40]. This has been suggested to cause a rapid cooling of the northern hemisphere, with glacial advances commencing within about 12 years in Greenland [40]. It was called the "bipolar see-saw" by Broecker [41–43]. After this, a period of increasing expansion of cold glacial conditions takes place for about 85 ka, with the sea level falling as more water is stored as ice on land. The cold periods are punctuated by minor warming episodes about every 23 ka. It finally ends when the Arctic air mass reaches an extent such that the warming caused by the change in tilt of the axis of the Earth (precession) causes the Arctic air mass to retreat with its main flow changing from Paths II and III to Path I (Figure 1), i.e., from the northern paths to that centered on southern British Columbia (Figure 1). This produced deglaciation and an Interglacial warm period lasting 10–15 ka. The Milankovitch cycles, specifically the eccentricity and the precession cycles, are believed to control the system [5,6].

The source of the thermohaline water is regarded as being due to intense evaporation of surface water into the cold, dry Siberian air over the open parts of the Arctic and North Atlantic Oceans, leaving behind denser, more saline brine that sinks to the ocean floor. Although some of it moves south continually, it discharges south at irregular intervals and in varying volumes. The whole 100 ka cycle of glacial and interglacial events has been referred to as "the climatic cycle", although its periodicity is only constant in the Northern Hemisphere if the geography of the land and sea remains the same [44].

*3.5. The Intergovernmental Panel on Climate Change (IPCC) Proposal*

The IPCC is sponsored by the United Nations Organization and consists of selected climate scientists from several different countries. Their proposal in 1988 [17] is that human activities have resulted in increased atmospheric carbon dioxide, causing an increase in global temperature that overrides all other causes. It is assumed that the increase in

atmospheric carbon dioxide since the beginning of the industrial revolution is the cause of the warming [17]. This is not consistent with studies involving changes in temperature in rural areas of the northern hemisphere [18,19] or in much of the southern hemisphere. It is true that it is a greenhouse gas, but it only affects a small range of long-wave reradiation from the surface of the Earth. The latter has a much wider range of wavelengths in its reradiation that depend on the temperature of the radiating surface. The increase in this gas is measured primarily at a single station at the summit of Mount Kea on the island of Hawaii, and the increase parallels the change in air temperature at that station since about 1900 A.D. It is generally assumed that it does not vary significantly around the globe except for minor seasonal changes. No consideration is given to the fact that as the water in the oceans warms, the carbon dioxide dissolved in it decreases in solubility, and degassing takes place. This degassing from the oceans is slow and matches the increase in temperature of the upper 2000 m of the North Atlantic Ocean, at any rate for the data for that location since 1910. The warming appeared to precede increasing carbon dioxide concentrations during the last deglaciation at 24 sites around the world during the last deglaciation [43], but this was the result of comparing surface water temperature with the total carbon dioxide degassed from the entire water column at each site. The relationship of carbon dioxide to atmospheric air temperature has been widely discussed [44], and it has been shown that temperature changes precede changes in atmospheric carbon dioxide in the case of Antarctic cores [45,46].

Payet and Holmes provide summaries of some of the main arguments questioning the validity of the IPCC theory [47,48], while Christy has testified before the U.S. Congress that the mathematical models used by the IPCC do not match the real-world observations [49]. The theory has been embraced by governments, research workers who saw it as a means of obtaining research grants, commercial firms who saw the possibilities of new work, environmentalists, and the press since it was a simple explanation that could easily be understood by the public, but it has been severely criticized by a substantial number of experienced scientists. For example, there have been over 75,000 comments published on ResearchGate concerning the relationship between seawater temperature and increasing atmospheric carbon dioxide. Many are not very chivalrous!

An obvious problem is found when examining the map of the distribution of climate change (mean yearly air temperature) obtained by NASA from satellites (Figure 5). The main areas of warming are in Northern Canada and the Arctic, with lesser warming in the Sahara and the Australian Outback! Eastern China and Germany show no obvious warming. Obviously, this does not fit in with the main industrial centers in the world! Since atmospheric carbon dioxide is present in extremely low quantities and has a narrow band of wavelengths that it absorbs, it cannot possibly compete in effect with the much larger total solar radiation reaching the Earth's surface. It is a colorless, odorless gas with a molecular weight of 44 and is therefore mainly held down in the lower part of the atmosphere by gravity. Thus, models that assume that carbon dioxide rises to the outer portion of the atmosphere are unrealistic. Water, in all its phases, is a much more potent agent for moving heat around the globe.

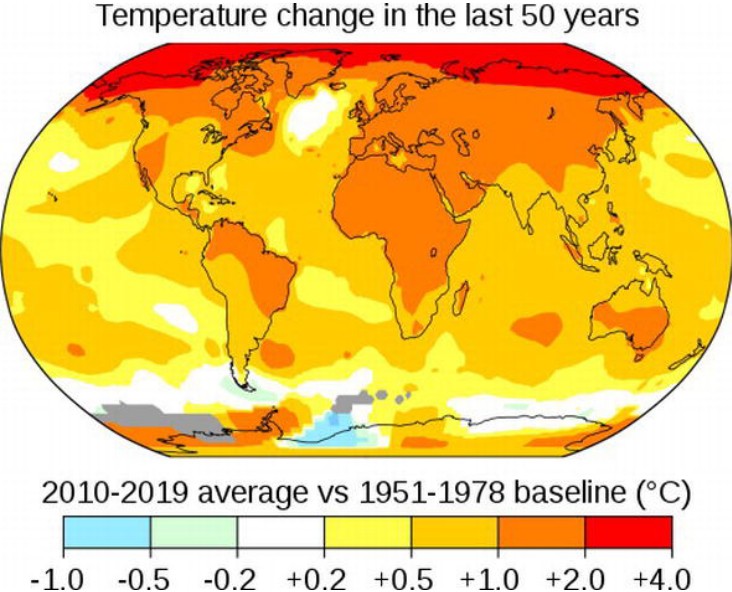

**Figure 5.** Map showing the measured mean annual temperature change around the world between 1951–1978 and 2010–2019 (NASA). The warming trend is not global and varies from being negative along the coast of Antarctica to over 4 °C around the Arctic basin.

The evidence for greater solubility of carbon dioxide in water at colder temperatures implies carbon dioxide moving from the air into the oceans during cold events [33] and can result in large quantities of the gas reacting with calcium ions to produce large amounts of calcium carbonate in the form of a calcareous, fine-grained limestone, e.g., during the Devonian and Carboniferous Periods. This implies that its abundance in the atmosphere is not entirely dependent on temperature. Both methane and carbon dioxide are chemicals that can and do take part in chemical reactions, whereas temperature is a measure of heat energy and cannot be created or destroyed. However, it can be changed into other forms of energy.

3.5.1. Ongoing Measurements of the Solar Energy Reaching the Surface of the Atmosphere

The basic difference between the IPCC proposal and the other ideas is the source of the additional heat being received in certain areas of the world. The obvious test is to measure the solar heat reaching the upper surface of the Earth's atmosphere. In 2015, the U.S. Government started collecting satellite measurements of the incoming radiation at the surface of the atmosphere over Salt Lake City, Utah (Figure 2), latitude 40° 26′ 20″ north, longitude 109° 57′ 30″ west from Greenwich.

Figure 6 shows the results obtained by the end of winter 2023, updated from Pangburn [50]. Shown in blue are the minimum winter temperatures compared with the preindustrial baseline, which are consistent with the precession cycle of Milankovitch commencing its decreasing mode of solar energy in the higher latitudes of the Northern Hemisphere in 2020. Thereafter, the winter temperatures at these sites decrease, indicating the commencement of a cooling trend that is likely to continue for the next 11.5 ka, based on the Milankovitch cycles. A corresponding warming trend should be occurring in the higher latitudes of the Southern Hemisphere. The current world record for cold is −98 °C, recorded in the Antarctic winter of 2018 [15], which is likely to stand for a long time since the change in the precession cycle should produce warmer winters there in the near future. In contrast, the winters in Western Canada and the southwest United States will be longer, colder, and have increasing precipitation.

This confirms the conclusion that the cold events involving glaciations are started by the 23 ka cycle of precession of the tilt of the earth's axis, not the 41 ka cycle as concluded by Milankovitch [6] and by Broecker [38,39]. The 41 ka cycle modifies the effects of

the precession cycle, as will other local geographic factors such as El Niño, ENSO, and Monsoons [51–53]. Carbon dioxide does not seem to be directly involved in the switch in winter climates in either hemisphere.

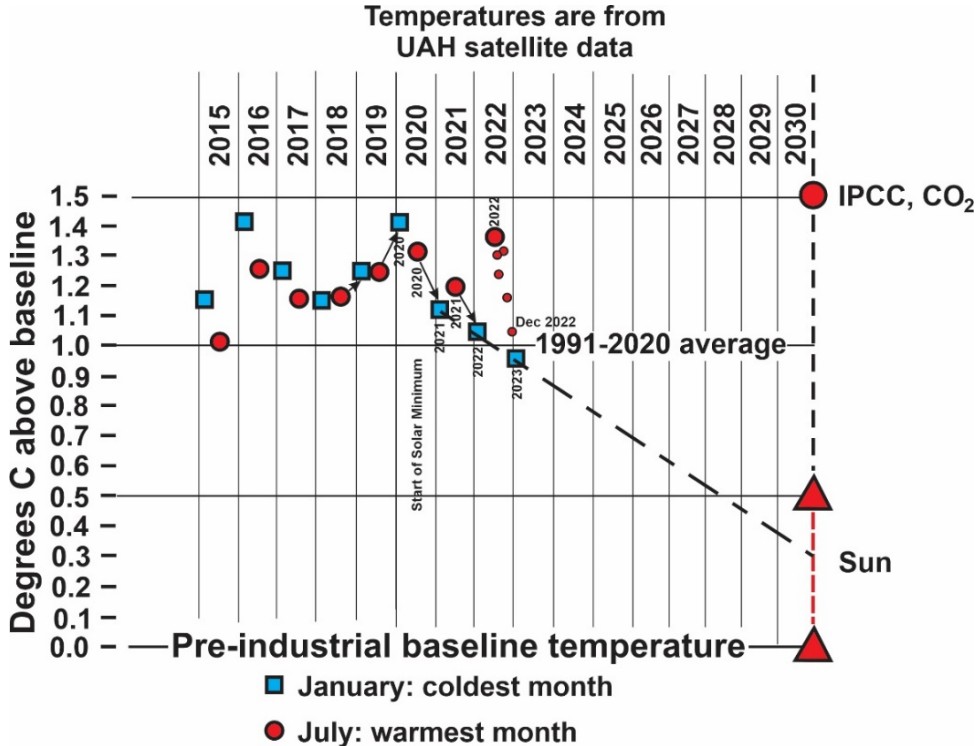

**Figure 6.** Comparison of incoming solar radiation for specific summer and winter months measured by satellites in the atmosphere over Utah from 2015 to 2023, showing the difference between actual totals by season and predicted IPCC $CO_2$ levels (modified from [52]).

The red values represent the maximum summer temperatures in Figure 2 recorded at the surface of the atmosphere. They differ from the blue values by showing ongoing increases since there are two summer maxima of air temperature between the Tropics of Capricorn and Cancer, corresponding to when the sun is directly overhead. The subtropical air masses originate in part in this zone, although the exact center of the zone from where the sun is overhead within the subtropical air masses depends on the local environment. The measured temperatures should start falling once the overhead sun has passed the center of the source of the subtropical air. Until then, the summer maxima will continue increasing, resulting in increasingly violent summer weather in Eastern Canada, the southeast, and central North America, as well as much of the rest of the world [9]. There will probably be stronger Monsoon rains throughout much of the world until the source areas of the storms undergo reduced solar heating.

The combined effects of the changes will have limited effects on the rising global sea level at first [48,49], followed by lower sea levels as water starts to be stored as ice on land in glaciers, in permafrost, and in some of the underground aquifers that have been seriously depleted by human activity.

## 4. Probable Changes in Landscapes over the Next 11.5 ka

If this climatic cycle follows a similar pattern of climate change to the Wisconsin glaciation, we can obtain some idea of the changes that will probably occur in the first 11.5 ka of this new glacial cycle in Western North America by examining what happened in the corresponding period for the previous cold event [10] (Figure 6). All three paths of movement of the Arctic air from Siberia are likely to be in action, although Path III will be dominant. This pattern is already establishing itself, with Paths II and III coalescing to

form enlarged Rossby Waves, while the increased precipitation along Path I when crossing the Rock Mountains has resulted in seven new avalanche paths in the Rogers Pass in 2001 A.D. and drought conditions followed by numerous wildfires in the Chinook area to the east in the southern Boreal Forest area of western Alberta that started in May 2023. These conditions also affect most of the areas of the Boreal Forest in Canada, and it remains to be seen how vegetation cover develops in these areas in the future.

The result of the interaction over South-Eastern and Central North America of the cold Arctic air of Path III with Subtropical tropical air coming up from Arizona and Mexico produces the large number of tornadoes and extreme weather in that area, together with the extremely hot, long summers. Possible complications could arise if the 41 ka cycle interferes with the climatic pattern or if tectonics or plate tectonics result in changes in the geography and topography of the area. Too low a concentration of atmospheric carbon dioxide could also wipe out the biota, leaving a barren landscape [39,43].

The Early Wisconsin glaciation mainly affected the north, and partial deglaciation took place after 11.5 ka, although it was both diachronous and incomplete. The ice sheets that survived the warming included the Keewatin, Baffin Island, Inuitian, Labradorian, and Cordilleran centers, but these centers did not always produce glacial advances at the same time, and their exact locations are still being debated [9], but they provide an indication of where ice sheets may form by about 11.5 ka. During the last glacial event (the Wisconsin Glaciation), the glaciers in southern British Columbia disappeared within another 20 ka after the first cooling 11.5 ka event.

By 11.5 ka, into the last glaciation, the TransCanada Highway and railway lines through the Rocky Mountains would have been destroyed by the Ice Cap in southern British Columbia, while the Northwest Passage would have disappeared under the Inuitian Ice Cap. An ice cap will develop over the highlands of Eastern Quebec and gradually spread south into the area of the Great Lakes. Obviously, there will be great impacts for humans, assuming they survive that long. An amount of 11.5 ka would permit at least 3000 generations, who would have to adjust to the new environments or move elsewhere. Thus, the changes in the environment are likely to be slow but inexorable. There will also be considerable changes in the ecosystems, together with changes in agriculture, land use, and the distribution of plants and animals. However, *Homo sapiens* has survived at least two previous cold events, although their numbers may have decreased considerably during these events. New plant and animal species evolve to make up for those that cannot adapt to the altered environments. New habitats will appear as sea levels change, causing the regrading of rivers moving upstream from the mouths of rivers, which creates new habitats for the biota. The actual shorelines would change substantially, though modified by isostatic changes [21,52].

Since the main areas of growth of the ice caps include Quebec and British Columbia, present-day electrical power generation in these areas will be greatly reduced, which will affect the use of electricity as the main future power source as currently planned. It will have to be replaced by other sources, such as oil or gas. Solar power will be reduced by the changes once the incoming solar radiation starts to decrease. Wind power will increase but may be difficult to safely control as the weather becomes more extreme. Any atomic power plants must be kept away from the probable paths of glaciers to prevent radioactive contamination of the environment. Arulich provides an excellent critique of some of the proposed changes in sources of energy suggested to replace coal and natural gas, suggesting that these are attainable solutions [53]. We will probably need multiple energy sources to survive in the future.

*Population Changes*

Neanderthal man had become extinct in Europe by c.1200 B.C., so the population figures for the succeeding centuries refer to *Homo sapiens* (Figure 7). The striking feature is the enormous growth in numbers since the end of World War II. According to the UN, this is expected to increase to 11–15 million people by the end of this century if nothing

interferes too seriously. Obviously, it will be difficult to move that number of people to warmer climates during the forthcoming cold events, so there will have to be some major changes to avoid major losses of life in the future. Even now, the population is having difficulties sustaining itself without exhausting the available natural resources on Earth. Current migration from tropical countries and islands to northern countries such as North America and Europe, which is perceived as a way to riches, represents "jumping out of the frying pan into the fire". There are already too few people there to sustain the life the immigrants are looking for, and climatic change is making life much more difficult for those already living there. Climatic change does not look as though it will improve the situation.

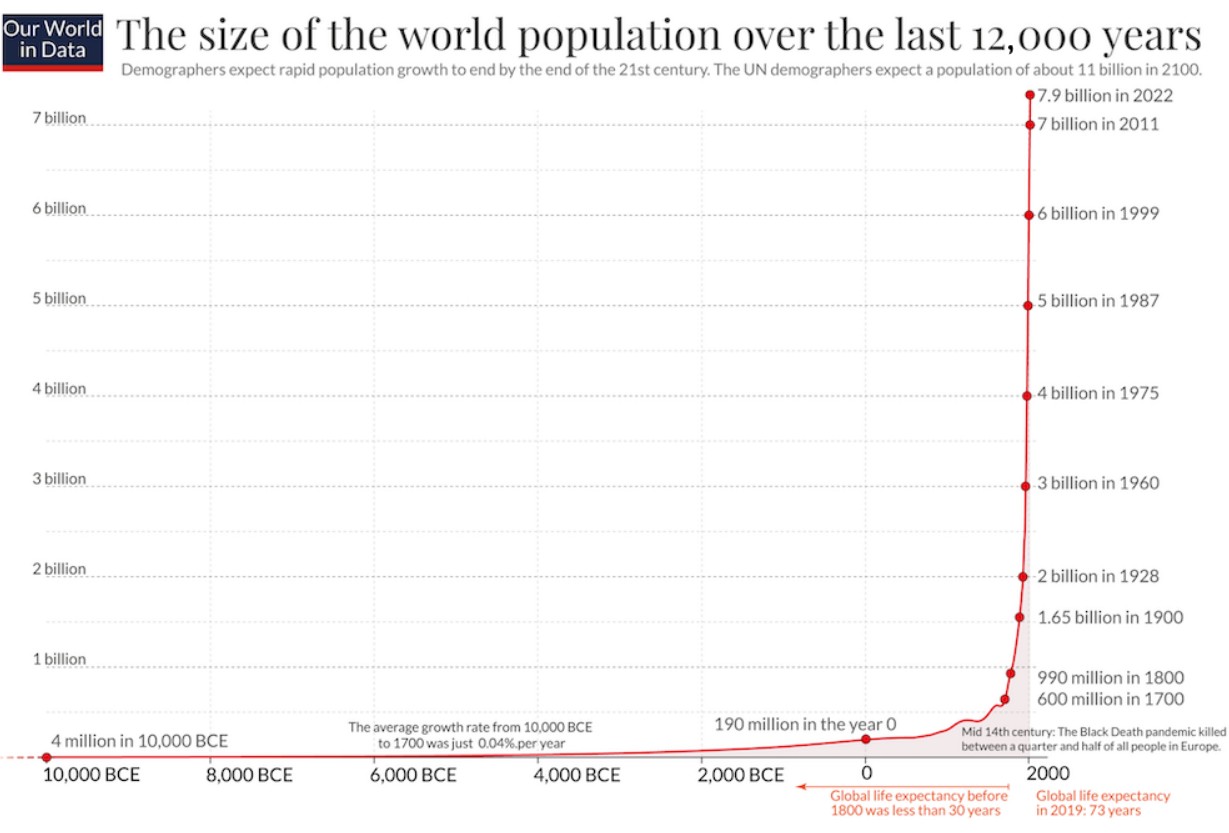

**Figure 7.** Changes in world population over the last 12,000 years (UN data).

### 5. Public Policy, Economies, and Financial Decisions

Climate change has a considerable effect on the finances and future economic problems faced by countries [54]. The current policies seem to be based primarily on the ideas promoted by the IPCC. Unfortunately, these are not soundly based on the science regarding the climate challenges facing Mankind. An example is the use of climatic data from ground temperature cables along the Trans Alaska Pipeline route instead of those from the Class 1 weather stations that are run by the US Government. Heat loss from the pipeline causes the air temperatures adjacent to it to be significantly higher than elsewhere. The IPCC used the pipeline data to claim that all of NW North America was suffering from climate warming, but the Class 1 US weather stations do not show any such long-term changes. Instead, they are controlled by the North Pacific Oscillation.

There is considerable confusion among the Public as to whether we are facing global warming based on the effects of industrialization affecting the whole world or climatic change caused by external factors such as the Milankovitch cycles modified by local Geographic factors. Yet a third group of people are climate change deniers. This is the result of a failure to teach the science of climate change in the Education system and in

news reports. Although the Milankovitch cycles are widely accepted [55–57], they are ignored by the IPCC.

The acceptance of decarbonization as suggested by the IPCC has resulted in governments blindly interfering with the coal and gas industries to switch to alternative energy sources. As noted above, the successful elimination of atmospheric carbon dioxide would eliminate life on Earth as we know it [49,50]. Atmospheric carbon dioxide is not the cause of climatic change, whereas the actual causes are connected to the Milankovitch cycles, specifically the 23 ka cycle, as explained in Section 3.5.1 above. When the tilt of the axis of the Earth changes, this produces changes in the summer and winter insolation. Initially, the winters become colder and longer, whereas the summers become warmer and longer until the overhead sun at midday moves south of the source area of the Subarctic air mass. Thereafter, the summers will also cool. At present, we are in the in-between period when the winters are cooling but the summers are still getting hotter. This brings about markedly changed climate patterns involving both the temperature and moisture regimes.

Some of the changes in North America include drought east of the continental divide, hot conditions, thunderstorms, and tornadoes. Lightning strikes ignite forest fires in the Boreal Forest and on the grassy plains. West of the continental divide, periodic floods due to monsoon-like rivers of rain from the Tropics can cause the inundation of floodplains such as the Fraser Valley, etc. In the southwest of the United States, drought conditions result in the drying up of reservoirs; e.g., behind the Hoover Dam, interfering with irrigation and power generation. Summer temperatures become extremely hot, both there and in southern and eastern North America. The Arctic Front, separating the Arctic air from the Subarctic air, generates extreme weather due to the increased differences between the two air masses, resulting in extreme flooding as well as large tornadoes. Similar changes occur elsewhere, e.g., the Saharan dry, hot air mass expands across the Alps into Germany and south into East Africa, causing drought conditions, while the Indian and East China monsoons become more extreme.

All these changes produce difficulties for people who are used to more benign weather and are causing enormous concern. The Alberta Government is trying to curb increases in insurance rates, but at least one insurance company has given notice that it will not insure Albertans next year because of the extremely large claims submitted to them dealing with the damages caused by extreme weather. Since carbon dioxide is not the cause but is needed by plants for photosynthesis, pumping this gas underground at great cost should cease, along with an end to the carbon tax. This would provide help to the taxpayers who are having difficulty paying rent, the cost of housing, and food bills. Fiscal policies induced by the views of the IPCC have severely damaged the economies of many countries, and this needs correction.

## 6. Conclusions

Enough theories have been tried and tested so that we now have a much better idea of how the climatic cycle works. The cycle commenced as soon as the Earth cooled down and is closely related to the main source of heating coming from the Sun and the Milankovitch cycles [5,6]. The Sun has been steadily warming since the beginning of the Earth's history [31]. If the Astronomers are correct, this heating will continue until the Sun becomes a Red Star and swallows up the inner four planets one by one, possibly starting about 5 Ma in the future. This increase is superimposed on the 23 ka, 41 ka, and 100 ka cycles resulting from the relative positions and movements of the Sun and the Earth. Carbon dioxide is a gas that is of fundamental importance to life as we know it. If its concentration in the atmosphere becomes too low, the bulk of the living things on the surface of the Earth will die, and the surface will become as barren as the other planets in the solar system [31,43]. There seems to be no connection between carbon dioxide and the temperature of the Earth [14,19,28,29,43–46]. Accordingly, the policies used by policymakers need to be changed to eliminate the burial of carbon dioxide underground, not provide large sums of public money to foreign firms to build battery factories, and

realize that we will still need the oil and gas industry in the future. It is an essential part of the economy, and in the future, any necessary pipelines should not be seriously considered. The gas tax should be eliminated.

The climate of the Earth is driven by the uneven solar heating of the surface of the Earth and the movements of the excess heat in the tropics towards the cooler polar regions, primarily by the movements of ocean currents, modified by the movements of air masses. The rotation of the Earth results in the Coriolis force causing fluids to rotate in a clockwise direction in the northern Hemisphere and in an anticlockwise direction in the southern Hemisphere. It also results in an eastward movement of the air masses around the Poles of the Earth (Figure 1). Oceans make up 70% of the surface of the Earth, and the thermal properties of water result in ocean currents being the primary method of transporting heat towards the poles, aided by hurricanes. The circular shape of Antarctica prevents the direct transport of heat to Antarctica, in contrast to the heating of adjacent land areas of the Northern Hemisphere via the North Atlantic Ocean. The excess heat in the North Atlantic Ocean causes intense evaporation of sea water, producing dense, deep-water thermohaline masses that periodically move south to the colder water circulating around Antarctica, thus causing a periodic return flow of cold Antarctic surface water to the North Atlantic.

Finally, it should be noted that the expansion of the cooling continues through four 23 ka cooling events before the warming phase of the last 23 ka cycle triggers excess pressure in the cold Arctic air mass, resulting in the resurrection of the Arctic Air escaping south along Path II, relieving the pressure from the Subtropical Air mass from the south. In this way, the excess mass of Arctic air is moved south and converted into Subtropical air, allowing deglaciation to take place and the commencement of the next short warm event (Interglacial) in the northern hemisphere.

**Funding:** This research received no external funding.

**Institutional Review Board Statement:** Not applicable.

**Informed Consent Statement:** Not applicable.

**Data Availability Statement:** Not applicable.

**Conflicts of Interest:** The author declares no conflict of interest.

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
