# Peer review of "Comparison of Recently Proposed Causes of Climate Change"

_atmosphere, doi:10.3390/atmos14081244_

Round 1
Reviewer 1 Report
The idea and execution of the paper are both very interesting, timely, and contribute to the recent academic debate about climate change. Thus, I have few comments that should be addressed.
1- The structure of section 1 should be revised: Why having one subsection 1.1, and several sub-subsections 1.1.2, etc... and no subsection 1.2?
2-It would be very nice if the authors can refer in a brief way to the impact of climate change and climate risk on financial markets and the investment landscape to widen the scope of the audience and ultimately the odds of being cited when the paper gets published. They can consider the following studies:
https://doi.org/10.1111/joes.12551; https://doi.org/10.1016/j.frl.2022.102740 ; https://doi.org/10.1016/j.jclepro.2023.137594 ;https://doi.org/10.1007/s11356-023-26947-w.
3- It is not clear what do you mean by "The problem was partly solved by making a histogram of all the available dated glacial deposits and associated permafrost features in North America".
4-In the conclusion section, the authors should elaborat2 eon the implications of the analysis for the policymakers.
5-Try to make the figures more self-explanatory.
See above.
Author Response
- Done.
- Discussed in a new section.
- Reworded.
- Done.
- Done.
Reviewer 2 Report
The author would include a few schematic figures showing the relationships among different paths mentioned in the text. The author would also include some discussions of the roles played by natural climate modes, such as those on interannual (ENSO), decadal (PDO) and interdecadal (AMV) time scales.
The author would include a few schematic figures showing the relationships among different paths mentioned in the text. The author would also include some discussions of the roles played by natural climate modes, such as those on interannual (ENSO), decadal (PDO) and interdecadal (AMV) time scales.
Author Response
Several extra figures have been added to support the ideas discussed in the text.
Reviewer 3 Report
The issue of addressing climate change is very important at the moment. The work submitted for review presents a very interesting overview of the phenomena that influence climate formation. It does not, however, provide an answer as to how humans should behave in the current reality. Certainly not to wait.
In the work, there are a great many themes that could be developed or summarised more than once (e.g. human influence on climate change, or current measurements of solar energy; changes in the landscape, what follows). Whether these results can be globalised. It is also not clear what is comparable to what, as suggested by the title of the publication (perhaps a subtle change in the title of the publication) and what is the main objective of the publication, as this one was not presented.
However, the overall assessment of the manuscript is that it is very well evaluated and, in my opinion, can be published with small additions.
Author Response
Small additions have been added although I did want to make the paper too long.